# Evaluation of Transgenerational Effects of Sublethal Imidacloprid and Diversity of Symbiotic Bacteria on *Acyrthosiphon gossypii*

**DOI:** 10.3390/insects14050427

**Published:** 2023-04-29

**Authors:** Yindi Wei, Yue Su, Xu Han, Weifeng Guo, Yue Zhu, Yongsheng Yao

**Affiliations:** College of Agriculture, Tarim University, Aral 843300, China; 10757213007@stumail.taru.edu.cn (Y.W.); xjsuyue@163.com (Y.S.); 120050099@taru.edu.cn (X.H.); 120040037@taru.edu.cn (W.G.); zyyinsect@163.com (Y.Z.)

**Keywords:** *Acyrthosiphon gossypii*, imidacloprid, hormesis, transgenerational effects, symbiotic bacteria

## Abstract

**Simple Summary:**

*Acyrthosiphon gossypii* is a sap-sucking pest that causes direct cotton damage by sucking sap and indirect damage by honeydew contamination and induced fungal growth. Chemical pesticides are widely used to combat the damage caused by *A. gossypii*. However, the escalating use of pesticides has had a notable impact on the resurgence and resistance of pests. In addition, microbes that live in insects as symbionts have recently been found to protect their hosts against toxins. In this study, we determined the toxicity and sublethal effects of the neonicotinoid insecticide imidacloprid on *A. gossypii* and further determined the effects of imidacloprid on symbiotic bacteria. The sublethal concentrations of imidacloprid had a negative effect on the reproduction and longevity of the G0 generation of *A. gossypii*, increased the population growth parameters of the G1 and G2 generations, but not those of the G3 generation, and altered the abundance of symbiotic bacteria. Our results revealed the transgenerational and sublethal effects of imidacloprid on *A. gossypii*, which may have implications for the endosymbiont-insecticide of the major pest.

**Abstract:**

Symbiotic bacteria and hormesis in aphids are the driving forces for pesticide resistance. However, the mechanism remains unclear. In this study, the effects of imidacloprid on the population growth parameters and symbiotic bacterial communities of three successive generations of *Acyrthosiphon gossypii* were investigated. The bioassay results showed that imidacloprid had high toxicity to *A. gossypii* with an LC_50_ of 1.46 mg·L^−1^. The fecundity and longevity of the G0 generation of *A. gossypii* decreased when exposed to the LC_15_ of imidacloprid. The net reproductive rate (*R*_0_), intrinsic rate of increase (*r_m_*), finite rate of increase (*λ*), and total reproductive rate (*GRR*) of G1 and G2 offspring were significantly increased, but those of the control and G3 offspring were not. In addition, sequencing data showed that the symbiotic bacteria of *A. gossypii* mainly belonged to Proteobacteria, with a relative abundance of 98.68%. The dominant genera of the symbiotic bacterial community were *Buchnera* and *Arsenophonus*. After treatment with the LC_15_ of imidacloprid, the diversity and species number of bacterial communities of *A. gossypii* decreased for G1–G3 and the abundance of *Candidatus-Hamiltonella* decreased, but *Buchnera* increased. These results provide insight into the resistance mechanism of insecticides and the stress adaptation between symbiotic bacteria and aphids.

## 1. Introduction

Cotton, as an important economic crop, is widely planted all over the world [1]. In China, Xinjiang, cotton area is the biggest production base of high-quality cotton, with an area of 24,970 hm^2^ and a total production of 53,910 tons in 2022, accounting for 83.2% and 90.2%, respectively. *Acyrthosiphon gossypii* Mordviiko (Hemiptera: Aphididae) is an important cotton pest in Xinjiang province, China. This species is widely distributed in Central Asia and India, through southern Ukraine and the Middle East to the Mediterranean and Atlantic [2], causing economic losses to legumes and cotton crops [3]. In addition, it also occurs in Malvaceae, Cruciferae, Tribulus, Compositae, and Rosaceae plants [4]. In China, *A. gossypii* is only distributed in Gansu and Xinjiang provinces [4]. This pest causes damage directly by sucking the juice of the young leaves and tender shoots, as well as indirectly by secreting honeydew, which may induce cotton sooty mold and affect the quality of the cotton [5]. Currently, chemical pesticides are widely used as the main measures to control cotton aphids. Due to a great quantity of chemical pesticides being used for a long time, cotton aphids have developed different levels of resistance and resurgence [6,7,8], which has become a key challenge for chemical pesticides. These characteristics have drawn increasing attention to the safety and application of pesticides.

Neonicotinoids are a major class of insecticides, which have been successfully used for the control of sucking insect pests since the 1980s. Imidacloprid is the first neonicotinoid insecticide and has stomach toxicity, inhalation, and contact insecticidal activity. Imidacloprid has been proven effective to control pests with piercing-sucking mouthparts and is widely used in more than 140 crops in agricultural production [9,10]. Imidacloprid acts on acetylcholine receptors of target insects to cause paralysis to death [11] and is lowly toxic to fish, amphibians, reptiles, birds, and mammals [12]. However, imidacloprid is highly toxic to nontarget organisms. These include *Ceratomegilla undecimnotata* [13], *Hippodamia variegata* [14], *Trichogramma cacoeciae* [15], and other natural enemy insects and pollinators [16,17]. Especially, over-dependence on pesticides has resulted in the development of resistance and resurgence of pests such as *Sitobion avenae* [18], *Rhopalosiphum padi* [18], and *Nilaparvata lugens* [19], and pest prevention and control is becoming increasingly difficult.

In addition, pesticides cause significant death of target pests, and low-lethal concentrations of chemical insecticides that have degraded over time could also exhibit sublethal effects to exposed arthropods due to environmental factors in agricultural ecosystems [20,21]. The exposure of insects to this concentration of pesticides affects their lifespan, development, survival, and fecundity [22,23]. At a certain concentration, the fecundity and longevity of surviving individuals may increase [18], but in some cases, these surviving individuals can inhibit fecundity and longevity [24]. Stimulatory effects by the sublethal dose of insecticide are generally known as hormesis, which is an adaptive manifestation of the continuous evolution of organisms under environmental stress, and these hormesis effects could enable organisms to better respond to subsequent exposure to higher levels of stimuli [25] and be linked to the pest resurgence under field conditions.

Hormesis induced by sublethal insecticides is the change at the population level caused by the stress response of the individual organism in the population, which is related to the outbreak of insect pests and insecticide resistance [26]. The physiological and/or biochemical changes in organisms after sublethal concentrations of pesticides can explain the enhanced tolerance or resistance of organisms [27,28]. Previous studies on insecticide resistance focused on the overexpression of these critical detoxification genes which can also explain the enhanced tolerance in insects [29]. For instance, the overexpression of P450 CYP6AY1 leads to an efficient metabolism of imidacloprid and induces resistance of *N. lugens* to imidacloprid [30]. Moreover, insecticide stress remarkably affects the symbiotic bacteria in insects [31]. *Bacillus cereus* in the intestinal tract has been shown to degrade the resistance of *Plutella xylostella* to pirimicarb [32]. The secondary symbiont *Hamiltonella* plays an important role in the wheat aphid *Sitobion miscanthi* and increases its resistance to acetamiprid, chlorpyrifos, cyanobenzamide, and imidacloprid [33]. Together, these studies suggest that symbiotic bacteria mediate insecticide resistance in insects. Nevertheless, how sublethal concentrations of imidacloprid exposure can affect the biological traits of the population and the symbiotic bacteria in *A. gossypii* is still unclear.

A life table was constructed to evaluate the sublethal effects of imidacloprid on the biological traits and population growth parameters of *A. gossypii*. In addition, to explore the potential association of sublethal effects of imidacloprid, 16S rDNA sequencing analysis was used to study the changes in symbiotic bacteria in the *A. gossypii* population under imidacloprid stress. The results of this study provide a reference for understanding the chronic effect of imidacloprid stress on the symbiotic bacteria of *A. gossypii*.

## 2. Materials and Methods

### 2.1. Insect Rearing and Insecticide

The *A. gossypii* used in this study were collected from Tarim University Agricultural Experimental Station cotton fields where cotton is not sprayed with any pesticide during the whole growth period (Aral, Xinjiang province) on 10 June 2021. They were reared on pot-grown cotton (Xinluzhong No. 67) as the host under the conditions of temperature 22 ± 1 °C, relative humidity 60% ± 5%, photoperiod L//D = 16 h//8 h, and no contact with any pesticide in the climate chamber. After four generations of propagation, the experimental population strains were established. Imidacloprid (99% purity) was supplied by Taihe Pharmaceutical Co., Ltd. (Ji’an, China). Triton X-100 was purchased from Solebold Technology Co., Ltd. (Beijing, China). 

### 2.2. Bioassay of Imidacloprid Toxicity to A. gossypii

Under laboratory conditions, the toxicity of imidacloprid to *A. gossypii* was determined using the leaf dipping method [34]. Imidacloprid was added with a small amount of acetone until it was completely dissolved and then further diluted with distilled water containing 0.05% Triton X-100 to a gradient of 6 pesticide concentrations. The fresh cotton leaves without any contact with pesticides were soaked in the solution for 15 s and dried naturally. The 1.8% agar solution was poured into a petri dish (diameter 90 mm, height 20 mm). After solidification, the cotton leaves were attached to the agar, and 30 apterous adults (≤24 h) of the same size were placed in the petri dish in a constant temperature and humidity climatic chamber. Distilled water containing 0.05% Triton X-100 was used as the control. Each concentration was repeated three times, and the mortality rate was recorded after 72 h. The bioassay results were analysed using IBM SPSS Statistics 25.0 software to calculate LC_15_, LC_50_, and LC_90_ values and 95% confidence intervals.

### 2.3. Sublethal Effects of Imidacloprid on A. gossypii

To ensure the uniformity of the growth period of *A. gossypii*, more than 300 offspring nymphs produced on the same day were selected from the cotton leaves and fed continuously for 6 days as the G0 generation of the experimental population. The offspring of G0 adults were collected as the G1 generation, and so on. According to the toxicity determination results, the imidacloprid LC_15_ (0.54 mg·L^−1^) was selected as the sublethal concentration to treat *A. gossypii*. The fresh cotton leaves were immersed in the insecticide solution for 15 s, air-dried at room temperature, and placed in a corresponding number of apterous adults. After 72 h, the surviving individuals were transferred to a petri dish with leaves and 1.8% agar and raised individually in a petri dish in a climatic chamber. There were no less than 30 new nymphs (<24 h) in each group, with 10 replicates per treatment. The life table parameters of each individual were recorded every 3 h, including development time, daily reproduction, longevity, and other data. Fresh leaves and petri dishes were replaced every 4 days during the whole experimental period until all the G3 aphids died. The life table data of *A. gossypii* were imported into TWOSEX-MSChart software for analysis [35], and the net reproductive rate (*R*_0_), intrinsic rate of increase (*r_m_*), fecundity, finite rate of increase (*λ*), and mean generation time (*T*) were calculated. The mean and standard error of the experimental data were obtained by 100,000 random samplings using the bootstrap program. The obtained data were analysed using paired bootstrap test to examine the significant difference between the treatments and the control [36]. The survival, reproduction, and life expectancy curves were plotted using Sigmaplot 14.0 software.

### 2.4. Effect of Imidacloprid on the Symbiotic Bacteria of A. gossypii

The adults of *A. gossypii* were treated with imidacloprid LC_15_ at sublethal concentrations with acetone as a control. The apterous adults of the G1–G3 offspring were collected for 16S rDNA analysis. Each treatment was repeated four times.

Genomic DNA was extracted using a DNA Extraction Kit following the manufacturer’s instructions (Invitrogen, Carlsbad, CA, USA) under sterile conditions. The concentration and quality of extracted DNA were detected by NanoDrop2000 and agarose gel electrophoresis. Universal primers (343F: TACGGRAGGCAGCAG, 798R: AGGGTATCTAATCCT) of the V3-V4 region of the bacterial 16S rDNA gene were used for PCR amplification with Tks Gflex DNA Polymerase using genomic DNA as a template. The PCR products were assessed by electrophoresis and then purified by magnetic beads as a second round of PCR template for PCR amplification. The amplified DNA was assessed by agarose gel electrophoresis, purified by magnetic beads, and quantified by Qubit. The purified PCR product was mixed in equal amounts based on concentration and sequenced by OE Biotech Co., Ltd. (Shanghai, China). The raw sequence data obtained in this study were stored in the SRA database with the project registration number PRJNA943466. The raw data obtained by sequencing were stored in FASTQ format. Paired-end reads were then preprocessed using Cutadapt software to detect and cut off the adapter. After trimming, the paired-end reads were filtering low quality sequences, denoised, and merged, and -chimera reads were detect and cut off using DADA2 [37] with the default parameters of QIIME2 [38]. Finally, the software outputthe representative reads and the ASV abundance table. The representative sequences of each ASV were then selected for comparison with the Silva (version 138) database, and the species were annotated using q2-feature-classifier software annotation.

Based on the ASV abundance table, subsequent beta diversity (intersample) and alpha diversity (intrasample) analyses were performed [39]. The Chao1, ACE, Simpson, and Shannon indexes and Goods coverage in the alpha diversity analysis were used to generate dilution curves of different sample sizes to reflect the degree of species diversity in the biological environment. Based on the analysis results using the binary Jaccard distance matrix, the correlation between the control group and each treatment group was visualized by principal coordinate analysis (PCoA) [40]. LEfSe analysis was used to reveal the composition of different species in biological communities, and LDA > 2 [41]. SPSS 26.0 software was used for data entry correlation analysis and one-way ANOVA at the level of *p* < 0.05.

## 3. Results

### 3.1. Toxicity of Imidacloprid to A. gossypii Adults

The LC_15_, LC_50_, and LC_90_ of imidacloprid to adult *Acyrthosiphon gossypii* at 72 h were 0.54, 1.46, and 4.97 mg·L^−1^, respectively, and the 95% confidence intervals were 0.43–0.65, 1.27–1.68, and 3.94–6.79 mg·L^−1^, respectively (Table 1). The effects of imidacloprid on the population parameters and symbiotic bacterial community of *A. gossypii* were studied with LC_15_ as a sublethal concentration.

### 3.2. Sublethal Effects of Imidacloprid on the G0 Generation of A. gossypii

The fecundity and longevity of *A. gossypii* after 72 h of exposure to imidacloprid were evaluated (Figure 1). Compared with the control group, the LC_15_ of imidacloprid affected the fecundity and longevity of *A. gossypii*, and the number of G0 individuals in the treatment group decreased significantly.

### 3.3. Transgenerational Effects of Imidacloprid on the G1–G3 Generation of A. gossypii

The effects of sublethal concentrations of imidacloprid (LC_15_) on the growth, development, and reproduction of the G1–G3 generations of *A. gossypii* G0 are shown in Table 2. Compared with that of the control, the fourth instar nymph stage of the G1 generation was significantly shortened, and the adult pre-reproductive period (APOP) was significantly prolonged, but there was no significant difference in the G2 generation. In addition, longevity and fecundity increased significantly in the G1 and G2 generations, and the total preoviposition period (TPOP) was significantly shortened. There was no significant difference in the above life table parameters between the control and the G3 generation (Table 2).

The effects of sublethal concentrations of imidacloprid LC_15_ on the population growth parameters of the G1–G3 generation of *A. gossypii* are shown in Table 3. Compared with those of the control, the net reproductive rate (*R*_0_), intrinsic rate of increase (*r_m_*), finite rate of increase (*λ*), and gross reproduction rate (*GRR*) of G1 and G2 individuals were significantly increased after imidacloprid LC_15_ treatment. However, the mean generation time (*T*) decreased from 12.73 d and 13.11 d to 11.80 d and 11.97 d, respectively, and the population doubling time (*DT*) also showed a decreasing trend. There was no significant difference in the net reproductive rate (*R*_0_) and population doubling time (*DT*) between the G3 generation and the control (Table 3).

The age-specific survival rate (*l_x_*), age-specific fecundity (*m_x_*), and net maternity (*l_x_m_x_*) are shown in Figure 2. The *l_x_* curve of the G1–G2 generation of *A. gossypii* decreased in the late growth stage. Except for G3, the *lx* curve of the G1–G2 generation of the imidacloprid treatment group decreased later than that of the control. After imidacloprid treatment, the *m_x_* and *l_x_m_x_* of the G1 and G2 generations were higher than those of the control group, while there was a significant overlap between those of the G3 generation and the control.

The age-stage life expectancy (*e_xj_*) curve showed that the life expectancy of the G1, G2, and G3 generations of *A. gossypii* under imidacloprid LC_15_ stress was longer than that of the control group (Figure 3). The age-stage survival rate (*s_xj_*) curve was shown in Figure 4. Imidacloprid harmed the survival rate of three generations of nymphs and adults of *A. gossypii*. The *s_xj_* curve of imidacloprid effects on the G1 and G2 adults decreased later than that of the control, while the control group and the pesticide group of the G3 generation had no significant effect. In the age-stage reproduction value (*V_xj_*) curve (Figure 5), the reproduction value of *A. gossypii* increased with increasing age and reached a maximum at the adult stage. Under imidacloprid LC_15_ stress, the peak values of the G1, G2, and G3 treatments were 15.02 (9 d), 15.31 (11 d), and 15.45 (10 d), respectively, which appeared earlier than the control.

### 3.4. Sequencing Data

The 16S rDNA V3-V4 hypervariable region of *A. gossypii* was sequenced by the Illumina sequencing platform. A total of 79,877, 80,222, 79,337, and 79,825 raw reads were obtained from samples IG1, IG2, IG3, and CK, and each treatment was repeated four times (Table 4). Through quality filtering, noise reduction, stitching, and removal of chimaeric sequences, etc., at least 72,337 clean reads were obtained from each sample. These sequences were clustered into 109, 134, 131, and 143 ASVs in the IG1, IG2, IG3, and CK samples, respectively. The Goods coverage of sequencing data for all samples was estimated to be one (Table 4). When the depth of random sampling of the rarefaction curves of each sample (the number of sampled sequences) reached 40,000, the values of the Chao1 and Shannon indexes tended to be flat. It shows that the sequencing amount is representative (Figure 6a–c).

### 3.5. Diversity Analysis of Bacterial Populations

An alpha diversity analysis was used to estimate the diversity of species in the biological environment. It is represented by the Chao1, ACE, Simpson, Shannon, and other indexes (Table 4). The observed species values showed that the control group had the highest number of bacteria. The G0 generation of *A. gossypii* was treated with imidacloprid at the LC_15_, and the bacterial species of the G1–G3 generation were reduced. The Chao1 and ACE indexes estimated the actual number of species in the community, and the Simpson and Shannon indexes indicated the community diversity. Their trends were consistent with the observed species values, indicating that imidacloprid inhibited the diversity and species number of symbiotic bacteria in the G1–G3 generation of *A. gossypii*.

### 3.6. Taxonomic Composition of Bacteria in A. gossypii

The composition of symbiotic bacteria in *A. gossypii* changed greatly after treatment with imidacloprid at the LC_15_ for 72 h. A microbiome analysis showed that symbiotic bacteria were mainly distributed in Proteobacteria, Firmicutes, and Actinobacteria. Proteobacteria was the dominant phylum, accounting for 98.68%, 98.15%, 98.19%, and 98.10% in IG1, IG2, IG3, and CK, respectively. *Buchnera* and *Arsenophonus* were the dominant genera in the symbiotic community of *A. gossypii*. The average relative abundance of each sample accounted for 63.04% and 29.10%, respectively (Table A1).

### 3.7. Effects of Imidacloprid Treatment on Three Successive Generations of the Bacterial Community in A. gossypii

Beta diversity can reflect the community similarities and differences of different biological environments (different groups). In this study, PCoA was performed based on the binary Jaccard distance matrix to determine the differences between individuals or groups at the genus level. Each point in Figure 6d represents a sample, and the same colour represents the same group. The closer the sample in the same group was, the more obvious the distance from other groups was, indicating that the bacterial community structure was different and the grouping effect was good. Figure 6d shows that the bacterial community structures of the *A. gossypii* G1 and G2 generations and the control group were significantly different, but the control and IG3 samples had similar bacterial community structures.

At the family level, imidacloprid affected the diversity and abundance of symbiotic bacteria in the offspring (G1–G3) of *A. gossypii*. Under imidacloprid LC_15_ stress, the number of species with decreased abundances was greater than that with increased abundances. With the replacement of generations, the rate of decrease in the number of species slowed down in the G2 and G3 generations. The abundance of Lactobacillaceae, Bacteroidaceae, Muribaculaceae, and Lachnospiraceae decreased in the G1 generation, while the abundance of Morganellaceae and Rikenellaceae increased. The abundance of Lactobacillaceae, Bacteroidaceae, Muribaculaceae, and Rikenellaceae decreased in the G2 generation, while the abundance of Morganellaceae, Lachnospiraceae, Moraxellaceae, Prevotellaceae, and Oscillospiraceae increased. The abundance of Bacteroidaceae, Muribaculaceae, and Lachnospiraceae decreased in the G3 generation, while the abundance of Morganellaceae, Xanthomonadaceae, Lactobacillaceae, Flavobacteriaceae and Weeksellaceae increased (Figure 7a, Table A2).

At the genus level, the relative abundance of the top 15 symbiotic bacteria in the offspring of *A. gossypii* was significantly different after 72 h of treatment with sublethal imidacloprid concentrations (LC_15_). The symbiotic bacterial communities in the *A. gossypii* control group and the treatment groups were dominated by the primary symbiotic bacteria *Buchnera* and the secondary symbiotic bacteria *Arsenophonus*, accounting for 46–74% and 22–33%, respectively. After treatment with imidacloprid at the LC_15_, the relative abundance of *Buchnera* in the G1–G3 generations increased, and the abundance in the G2 generation increased by 28% compared with that of the control. The species number of *Candidatus_Hamiltonella* species in the three generations was lower than that of the control (Figure 7b and Figure 8, Table A3).

A LEfSe analysis revealed the composition of different species in two or more biological communities. The results showed that there were different biomarkers (LDA > 2.0) in the pesticide treatment group and the control group using sublethal imidacloprid concentration stress. Nine biomarkers were identified in all samples, and the biomarkers identified in the control group were *Pseudomonas*, *Lachnospira*, and Clade_Ia. Only one biomarker, *Succinivibrio*, was found in the G1 generation of the insecticide group. The biomarker *Buchnera* was found in the IG2 samples. *Sphingobacterium*, *Sulfitobacter*, *Yoonia_Loktanella*, and *Algitalea* were identified as biomarkers of IG3 (Figure 9).

## 4. Discussion

The results showed that imidacloprid had high toxicity to adult *A. gossypii*. The toxicity value (LC_50_) at 72 h was 1.46 mg·L^−1^, which is equivalent to the LC_50_ of imidacloprid reported by predecessors, and the LC_50_ value of 24 h exposure was 3.75 mg·L^−1^ [42]. In addition to the direct lethal effect, with the long-term continuous use of insecticides, random increases in dosage and application times, and other unreasonable phenomena, the toxicity of insecticides in the environment gradually decreased to sublethal doses over time after application, resulting in sublethal effects on insects [43,44]. Sublethal effects usually manifest as changes in insect growth and development, reproduction, ecological behaviour, etc. [45]. Excessive dependence on pesticides can easily lead to pest resurgence and an imbalance of ecosystem regulation [46]. In this study, sublethal concentrations of imidacloprid had significant effects on the life history traits and symbiotic bacteria of *A. gossypii*. Therefore, imidacloprid stress and symbiotic bacteria in *A. gossypii* affected the adaptability of organisms to stimuli.

Most pests are sublethally affected by direct exposure to low concentrations of insecticides, which usually have a negative impact on their biological traits. The results showed that the longevity and fecundity of G0 adults of *A. gossypii* decreased significantly after 72 h of imidacloprid treatment. This result is consistent with the sublethal effect of *Aphis gossypii* exposed to afidopyropen [24], imidacloprid [47], and acetamiprid [48]. Therefore, low concentrations of imidacloprid have an inhibitory effect on the current population growth of *A. gossypii*, and similar results were observed when *Aphis glycines* were exposed to acetamiprid [49].

The analysis of the life table parameters of this study showed that when the parent aphid (G0) was exposed to the sublethal concentration of imidacloprid LC_15_, the G1 and G2 nymphal and TPOP were shortened, while the longevity and fecundity were significantly increased (Table 2). These results indicate that the sublethal concentration (LC_15_) of imidacloprid had a stimulating effect on the G1 and G2 generations of *A. gossypii*. This hormesis of changes in the performance of organisms caused by sublethal doses of insecticides has been widely reported [50] and is often considered to be the main mechanism for the resurgence of pest populations [51]. When *A. glycines* Matsumura was stressed by a sublethal imidacloprid concentration (0.05 mg·L^−1^), the reproductive level and population growth increased [42]. Nitenpyram LC_20_ increased fitness for life table parameters and population growth parameters of *N. lugens* [28]. Similarly, low concentrations of imidacloprid stimulated reproduction in *Myzus persicae* [25].

The population growth parameters of the control and treatment groups of *A. gossypii* offspring (G1 and G2) observed in this study showed that the sublethal concentration (LC_15_) of imidacloprid affected the cross-generational population of *A. gossypii*. After imidacloprid stress, the *r_m_*, *λ,* and *GRR* of *A. gossypii* offspring (G1 and G2) increased significantly, while the *T* and *DT* were significantly shortened (Table 3). Therefore, imidacloprid promoted the growth of the *A. gossypii* population at low or sublethal concentrations. When the parent aphid (F0) was treated with thiamethoxam at a sublethal concentration (LC_15_), the longevity and fecundity of the *Aphis gossypii* progeny (F1) were significantly increased, and the biological traits *r_m_*, *λ*, *R*_0_ were significantly enhanced [52]. Sulfoxaflor has the potential to stimulate the reproduction of the F1 generation of *M. persicae* [53]. Flupyradifurone had similar effects on the F1 generation of *M. persicae* [54] and the offspring of *Aphis Craccivora* [55]. Therefore, low or sublethal concentrations of imidacloprid had significant hormetic effects on the offspring of *A. gossypii*.

Notably, the longevity and fecundity of the G0 generation were significantly inhibited when *A. gossypii* was stressed by sublethal concentrations of imidacloprid. The fecundity of its offspring (G1 and G2) increased significantly, and the population parameters *r_m_* and λ increased significantly. However, the life table parameters of the G3 generation treatment group returned to the same level as the control. This may reflect the adaptive trade-offs in the early generations of *A. gossypii* due to increased energy consumption required for survival and reproduction and increased fecundity (hormesis) [25,56].

As an important part of insect individuals, symbiotic bacteria form a mutually beneficial relationship with the host. They play an important role in insect nutritional function [57], reproductive regulation [58], and adaptability to biotic or abiotic factors [59,60]. Increasing evidence shows that many symbiotic bacteria of insects are related to the detoxification of chemical insecticides [33,61,62,63]. In this study, the microbial community in *A. gossypii* was explored by 16S rDNA Illumina sequencing analysis. After imidacloprid treatment, the bacterial community composition of *A. gossypii* changed significantly. PCoA showed that the bacterial community structure was disturbed in the offspring (G1–G2) and recovered in the G3 generation. The abundance of symbiotic bacteria also changed in the same way. In addition, after the parents were treated with imidacloprid at a sublethal concentration (LC_15_), the symbiotic bacteria of the offspring (G1–G3) of *A. gossypii* were mainly distributed in Proteobacteria, Firmicutes, and Actinobacteria, among which Proteobacteria was the dominant phylum (98.10–98.68%). At the genus level, both in the control group and the treatment group, *Buchnera* and *Arsenophonus* were dominant bacteria (Table A2). The results of this study are similar to those of the bacterial community composition of other aphids by imidacloprid [64,65]. The primary symbiont *Buchnera* plays a key role in the growth and development of insects, especially in the development of insect resistance [66,67]. Compared with the control, the relative abundance of *Buchnera* increased in three consecutive generations and increased to a significant level in the G2 generation after imidacloprid treatment. The results showed that *Buchnera* in the offspring of *A. gossypii* responded to pesticide stimulation after maternal exposure to imidacloprid for 72 h.

In addition, aphids contain one or more groups of bacteria called secondary symbionts. *Candidatus_Hamiltonella* is a common secondary symbiotic bacterium in aphids, and the infection rate is relatively high. Compared with the control group, the relative abundance of *Candidatus_Hamiltonella* in the imidacloprid treatment group decreased. Similar results were observed when *Sitobion miscanthi* was treated with chlorpyrifos-methyl, imidacloprid, cyantraniliprole, and acetamiprid for 24 h, and the density of the genus *Hamiltonella* decreased [33]. Our data suggest that *Hamiltonella* is involved in the adaptability of *A. gossypii* to imidacloprid and that the increase in the density of *Hamiltonella* may be beneficial to the development of resistance of *A. gossypii* to imidacloprid. Many other symbiotic bacteria may be contaminants caused by environmental factors, such as *Stenotrophomonas* and *Chryseobacterium*.

In summary, the LC_15_ of imidacloprid has a transgenerational effect on *A. gossypii*. After treatment with imidacloprid for 72 h, the fecundity and longevity of G0 generation individuals decreased. The *R*_0_ and *r_m_* of the G1 and G2 offspring were significantly increased, and the population grew rapidly and returned to normal levels in the G3 generation. In addition, imidacloprid affected the relative abundance of symbiotic bacteria and reduced the microbial diversity and species number of aphid offspring (G1–G3). Through LEfSe analysis, different biomarkers were identified in each group for three consecutive generations. The results of this study provide new ideas for the integrated management of *A. gossypii* and a theoretical basis for understanding the interaction between symbiotic bacteria and *A. gossypii*.

## 5. Conclusions

In the present study, we examined the adaptability of *A. gossypii* to imidacloprid by analysing the transgenerational sublethal effects of imidacloprid on population growth parameters and symbiotic bacterial communities. The results show that the LC_15_ of imidacloprid has a transgenerational effect on *A. gossypii*. The net reproductive rate (*R*_0_), intrinsic rate of increase (*r_m_*), finite rate of increase (*λ*), and total reproductive rate (*GRR*) of G1 and G2 offspring were significantly increased. The sequencing data showed that the diversity and species number of bacterial communities of *A. gossypii* for three consecutive generations (G1–G3) decreased, while the abundance of *Buchnera* increased. Our study revealed the transgenerational effect of sublethal concentrations of imidacloprid on *A. gossypii*, which will help to evaluate related studies on endosymbiont-insecticide resistance of *A. gossypii*.

## Figures and Tables

**Figure 1 insects-14-00427-f001:**
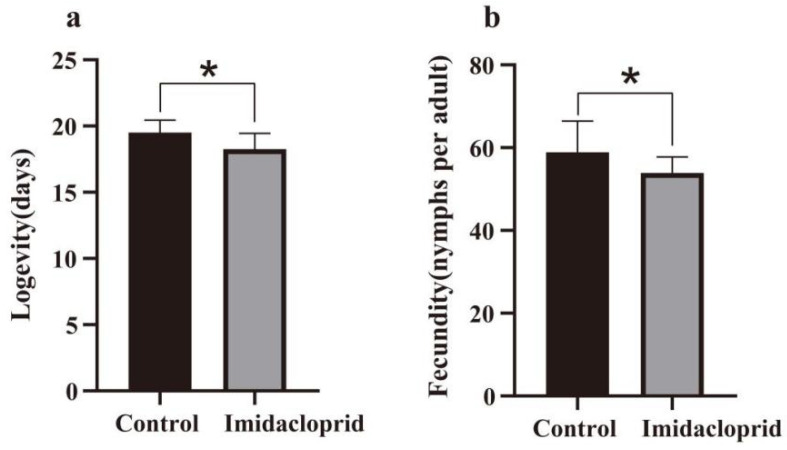
The longevity (**a**) and fecundity (**b**) of G0 generation *A. gossypii* exposed to the LC_15_ of imidacloprid for 72 h. Asterisk indicates a significant difference between the control group and the imidacloprid-treated group at the *p* < 0.05 level (*t*-test).

**Figure 2 insects-14-00427-f002:**
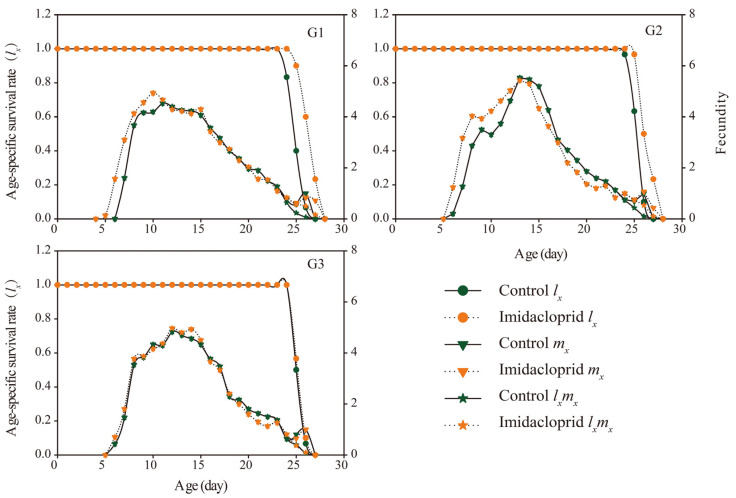
Age-specific survival rate (*l_x_*), age-specific fecundity of the total population (*m_x_*), and age-specific maternity (*l_x_m_x_*) of *A. gossypii*.

**Figure 3 insects-14-00427-f003:**
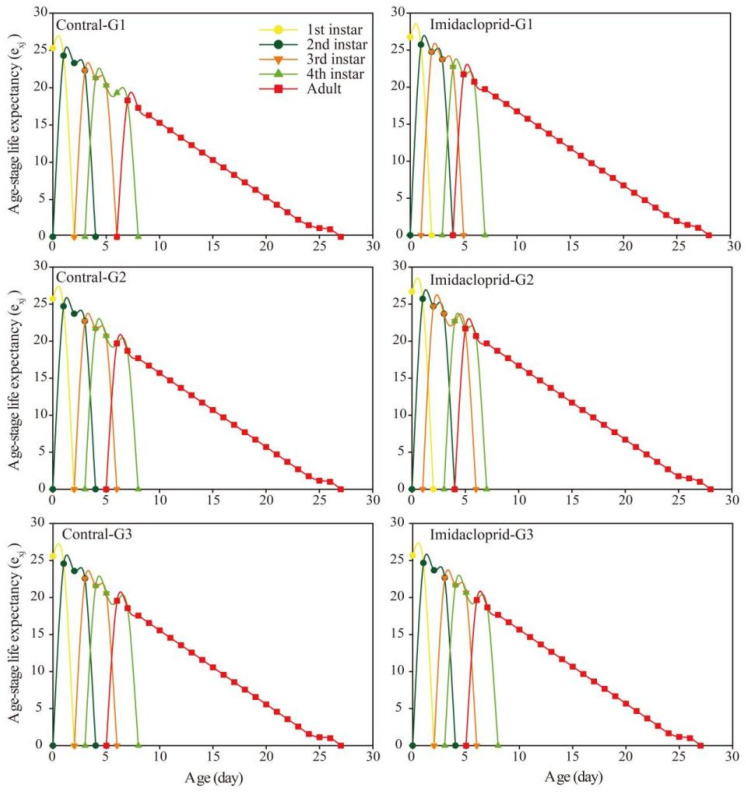
Age-stage life expectancy (*e_xj_*) of *A. gossypii*. under LC_15_ of imidacloprid exposure.

**Figure 4 insects-14-00427-f004:**
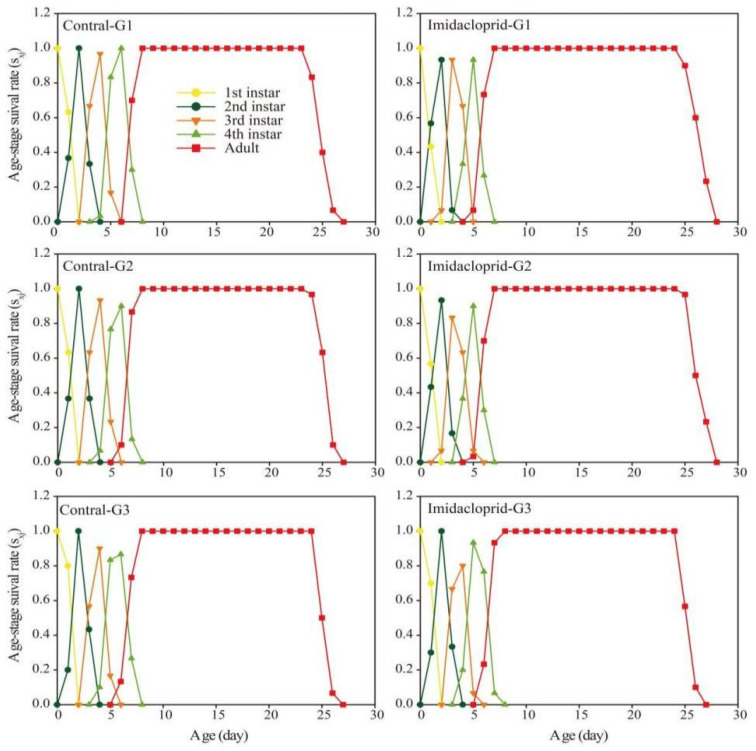
Age-stage survival rate (*s_xj_*) of *A. gossypii* under LC_15_ of imidacloprid exposure.

**Figure 5 insects-14-00427-f005:**
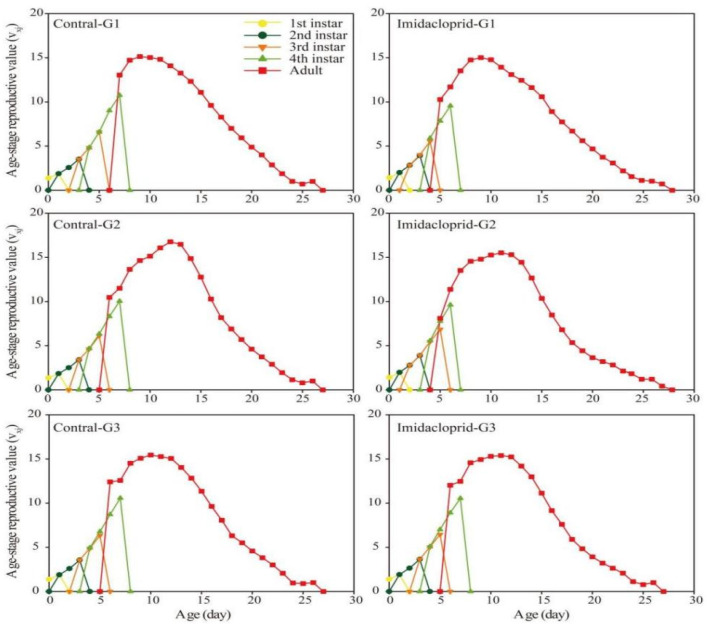
Age-stage reproductive value (*v_xj_*) of *A. gossypii* under LC_15_ of imidacloprid.

**Figure 6 insects-14-00427-f006:**
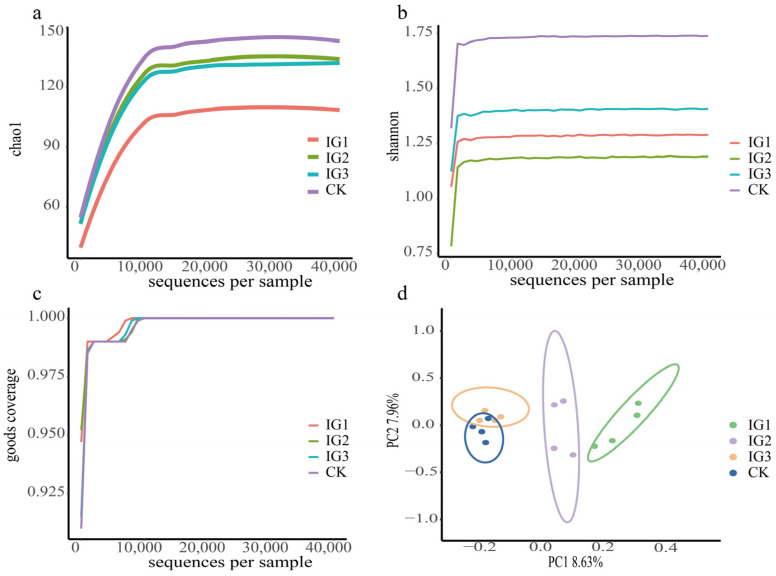
The Chao1 curve, Shannon curve and Goods coverage curve, and sample clustering analysis plot of bacterial communities in *A. gossypii* G1–G3. (**a**) Chao1 diversity index of all samples. (**b**) Shannon diversity index. (**c**) Goods coverage diversity index. (**d**) Principal coordinate analysis (PCoA) of G1–G3. The green, purple, yellow circles represent each generation (G1–G3) of imidacloprid group, respectively. The bule circle represent control group.

**Figure 7 insects-14-00427-f007:**
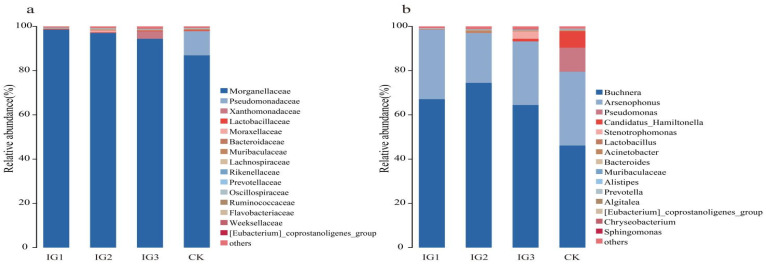
Changes of symbiotic bacteria of *A. gossypii* at different classification levels for three generations. (**a**,**b**) A histogram of relative abundance of the top 15 bacterial communities in *A. gossypii* G1–G3 after initial adult G0 exposure to imidacloprid LC_15_ for 72 h. The abundance values of ASVs in the treatment group and the pesticide group were analysed at the family and genus levels. The histogram is generated based on the data in Table A2.

**Figure 8 insects-14-00427-f008:**
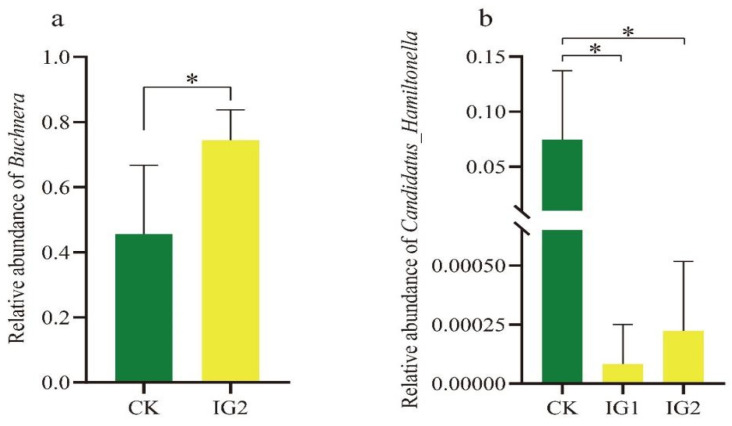
Comparison of relative abundance of two typical symbiotic bacteria in *A. gossypii* imidacloprid treatment group and control group. (**a**) an asterisk indicates a significant difference between the control group and the imidacloprid-treated group at the *p* < 0.05 level (*t*-test). (**b**) an asterisk indicates a significant difference based on one-way ANOVA followed by Turkey’s HSD multiple comparison test (*p* < 0.05).

**Figure 9 insects-14-00427-f009:**
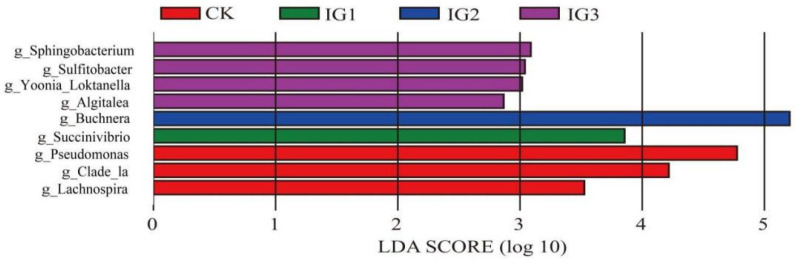
Identification of specific phylogenetic types at the genus level of bacteria between the imidacloprid-treated group and the control in three successive generations.

**Table 1 insects-14-00427-t001:** The toxicity of imidacloprid on *A. gossypii* adults.

Treatment	Slope ± SE	LC_15_ (mg·L^−1^) (95% CL)	LC_50_ (mg·L^−1^) (95% CL)	LC_90_ (mg·L^−1^) (95% CL)	R^2^	χ^2^ (*df*)
Imidacloprid	2.53 ± 0.21	0.54 (0.43–0.65)	1.46 (1.27–1.68)	4.97 (3.94–6.79)	0.97	4.08 (12)

SE: Standard error; 95% CL: 95% confidence limits; χ^2^: chi-square value; *df*: degrees of freedom.

**Table 2 insects-14-00427-t002:** Effect of imidacloprid LC_15_ on life history parameters of *A. gossypii* in the G1–G3 generation (mean ± SE).

Treatments		First Instar (d)		Second Instar (d)		Third Instar (d)		Fourth Instar (d)		Pre-Adult (d)		Adult (d)		APOP (d)		TPOP (d)		Longevity (d)		Fecundity	
G1	Control	1.63 ± 0.09	N	1.70 ± 0.08	N	1.80 ± 0.09	N	2.17 ± 0.08	*	7.30 ± 0.08	*	18.00 ± 0.12	*	0.07 ± 0.05	*	7.37 ± 0.09	*	25.30 ± 0.15	*	54.47 ± 0.89	*
	Imidacloprid	1.43 ± 0.09		1.57 ± 0.09		1.67 ± 0.09		1.53 ± 0.09		6.20 ± 0.10		20.53 ± 0.11		0.47 ± 0.12		6.67 ± 0.16		26.73 ± 0.17		59.83 ± 1.05	
G2	Control	1.63 ± 0.08	N	1.73 ± 0.08	N	1.80 ± 0.10	N	1.87 ± 0.10	N	7.03 ± 0.09	*	18.67 ± 0.13	*	0.43 ± 0.11	N	7.47 ± 0.13	*	25.70 ± 0.13	*	55.40 ± 0.91	*
	Imidacloprid	1.57 ± 0.09		1.53 ± 0.09		1.60 ± 0.09		1.57 ± 0.09		6.27 ± 0.09		20.43 ± 0.10		0.60 ± 0.11		6.87 ± 0.13		26.70 ± 0.16		59.07 ± 0.64	
G3	Control	1.80 ± 0.07	N	1.63 ± 0.09	N	1.63 ± 0.10	N	2.07 ± 0.08	N	7.13 ± 0.11	*	18.83 ± 0.14	*	0.17 ± 0.07	N	7.30 ± 0.13	N	25.57 ± 0.11	N	55.43 ± 0.89	N
	Imidacloprid	1.70 ± 0.08		1.63 ± 0.09		1.53 ± 0.09		1.97 ± 0.09		6.83 ± 0.10		18.43 ± 0.13		0.40 ± 0.10		7.23 ± 0.15		25.67 ± 0.12		55.83 ± 0.87	

Note: APOP, adult pre-reproductive period (days); TPOP, total pre-reproductive period (days); Fecundity, offspring per female. In the same generation, an asterisk in each column indicates that there was a significant difference between the control group and the imidacloprid-treated group at the *p* < 0.05 level (*t*-test), and N indicates no significant difference.

**Table 3 insects-14-00427-t003:** Intergenerational sublethal effects on the population growth parameters of *A. gossypii* in the G1–G3 generation (mean ± SE).

Treatments		*R* _0_		*r_m_*		*λ*		*T*		*GRR*		*DT*	
G1	Control	54.47 ± 0.89	*	0.31 ± 0.003	*	1.37 ± 0.004	*	12.72 ± 0.13	*	55.88 ± 0.96	*	2.21 ± 0.02	*
	Imidacloprid	59.83 ± 1.05		0.35 ± 0.006		1.41 ± 0.008		11.80 ± 0.18		60.77 ± 1.03		2.00 ± 0.03	
G2	Control	55.40 ± 0.91	*	0.31 ± 0.004	*	1.36 ± 0.005	*	13.11 ± 0.16	*	56.58 ± 0.91	*	2.26 ± 0.03	*
	Imidacloprid	59.07 ± 0.64		0.34 ± 0.005		1.41 ± 0.006		11.97 ± 0.15		59.95 ± 0.65		2.03 ± 0.03	
G3	Control	55.43 ± 0.89	N	0.32 ± 0.004	N	1.37 ± 0.006	N	12.66 ± 0.17	N	56.77 ± 0.91	N	2.19 ± 0.03	N
	Imidacloprid	55.83 ± 0.87		0.32 ± 0.005		1.38 ± 0.007		12.43 ± 0.19		57.01 ± 0.89		2.14 ± 0.03	

Note: *R*_0_, net reproductive rate (offspring per individual); *r_m_*, intrinsic rate of increase (d^−1)^; *λ*, finite rate of increase (d^−1^); *T*, mean generation time (days); *GRR*, gross reproductive rate (offspring/individual); *DT*, doubling time (days). Values in the table represent mean ± SE. In the same generation, an asterisk in each column indicates that there was a significant difference between the control group and the imidacloprid-treated group at the *p* < 0.05 level (*t*-test), and N indicates no significant difference.

**Table 4 insects-14-00427-t004:** 16S rDNA sequencing analysis of *A. gossypii*.

Samples	Raw Reads	Fitted	ASV_Counts	Observed Species	Chao1	Shannon	Simpson	ACE	Goods Coverage
IG1	79,877	73,689	109	108.28	108.57	1.29	0.50	108.81	1.00
IG2	80,222	74,207	134	133.50	133.95	1.19	0.41	134.21	1.00
IG3	79,337	72,337	131	130.30	130.65	1.41	0.50	131.38	1.00
CK	79,825	73,570	143	142.88	143.08	1.74	0.60	143.35	1.00

## Data Availability

The data presented in this study are available in the article.

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
