# Peer review of "Evaluation of Transgenerational Effects of Sublethal Imidacloprid and Diversity of Symbiotic Bacteria on Acyrthosiphon gossypii"

_insects, 2023, doi:10.3390/insects14050427_

Round 1
Reviewer 1 Report
This is a generally well-done study focusing on three types of imidacloprid effects on A. gossypii:
1) Toxicity
2) Sublethal effects – longevity, growth, fecundity
3) Symbiotic bacterial communities
The findings are well document and if sufficient importance and novelty warrant publication.
There are many edits (too many for me to make) needed to improve the presentation, but the science appears solid.
L11: combat the damage caused by
L14: we determined the toxicity and sublethal effects
L15: and also determined the effects of
L19-L20: G3 generations and altered the abundance of symbiotic bacteria.
L21: …on A. gossypii, which may have implications for the endosymbiont-insecticide resistance of the this major pest.
L24: adaptability is the wrong word – adaptation takes more than generations studied here. Just state something simple like “We studied the effects of imidacloprid on population growth parameters and symbiotic bacterial communities of Acyrthosiphon gossypii.”
L48: damaging cotton leaves and
Note: I stopped making editorial comments like the above after L48. Authors will need to edit for grammar themselves.
L110: “without contact with insecticides” means leaves were not treated in the field with insecticides?
This simulated foliar sprays. What about use as a systemic, where cotton leaves uptake the imidacloprid from the soil?
L117-119: State the response variable. Was it percent of the 30 aphids that died or were raw counts used.
L157-174: Put this analysis of genomic data into the same section as 2.4. Eliminate 2.4.1. and 2.4.2
L215: state that + / - are standard errors or standard deviations
L364 and throughout: Could longer life expectancy of treatment aphids be due to imidacloprid killing the naturally less tolerant individuals, leaving the naturally more resistant aphids alive. The control group comprises both tolerant and intolerant aphids.
Similar with fecundity. Imidacloprid or other insects perhaps weed out the genetically less fecund aphids (although unsure about genetic diversity in these aphids).
Author Response
Response to Reviewer 1 Comments
Dear editor and reviewers,
Thank you for offering us an opportunity to improve the quality of our submitted manuscript (insects-2324675). We appreciated very much the reviewers' constructive and insightful comments. We have revised the manuscript in which major changes are highlighted in red. These changes are summarized below following a point-by-point response to reviewer's comments. We hope the revised manuscript has now met the publication standard of your journal
Point 1: This is a generally well-done study focusing on three types of imidacloprid effects on A. gossypii: (1) Toxicity
- Sublethal effects-longevity, growth, fecundity
(3) Symbiotic bacterial communities
The findings are well document and if sufficient importance and novelty warrant publication.There are many edits (too many for me to make) needed to improve the presentation, but the science appears solid.
Response 1: Many thanks for your encouraging comments. Following all the comments and advice, we have now modified the manuscript accordingly.
Point 2: L11: combat the damage caused by.
Response 2: We have now revised the sentence (Line12).
Point 3: L14: we determined the toxicity and sublethal effects
Response 3: We have now revised the sentence (Line15).
Point 4: L15: and also determined the effects of.
Response 4: We have now revised the sentence (Line16-17).
Point 5: L19-L20: G3 generations and altered the abundance of symbiotic bacteria.
Response 5: We have now revised the sentence (Line19-20).
Point 6: L21: …on A. gossypii, which may have implications for the endosymbiont-insecticide resistance of the this major pest.
Response 6: We have now revised the sentence (23-25).
Point 7: L24: adaptability is the wrong word-adaptation takes more than generations studied here. Just state something simple like “We studied the effects of imidacloprid on population growth parameters and symbiotic bacterial communities of Acyrthosiphon gossypii.”
Response 7: Thanks for pointing out the mistake. We have now revised the sentence (27-29).
Point 8: L48: damaging cotton leaves and.
Response 8: We have now revised the sentence (Line 58).
Point 9: Note: I stopped making editorial comments like the above after L48. Authors will need to edit for grammar themselves.
Response 9: Thank you very much for the reviewer’s comments. Following your advice, we have carefully revised our paper.
Point 10: L110:“without contact with insecticides” means leaves were not treated in the field with insecticides?This simulated foliar sprays. What about use as a systemic, where cotton leaves uptake the imidacloprid from the soil?
Response 10: Sorry for the confusion. We have now revised the sentence.“without contact with insecticides”means that cotton is not sprayed with any pesticide during whole growth period. Following all the comments and advice from the reviewers, we have now modified the manuscript accordingly (Line126).
Point 11: L117-119: State the response variable. Was it percent of the 30 aphids that died or were raw counts used.
Response 11: In our article, 30 heads were placed in a petri dish as a group. There were no less than 30 new nymphs (<24 h) in each group, with 10 replicates per treatment. Moreover, within the group, the 30 aphids in the culture dish were themselves duplicates.
Point 12: L157-174: Put this analysis of genomic data into the same section as 2.4. Eliminate 2.4.1. and 2.4.2.
Response 12: We have revised it as your suggestion.
Point 13: L215: state that ± are standard errors or standard deviations.
Response 13: In the article, ± indicates standard errors. We have added standard errors in Table 2 and Table 3.
Point 14: L364 and throughout: Could longer life expectancy of treatment aphids be due to imidacloprid killing the naturally less tolerant individuals, leaving the naturally more resistant aphids alive. The control group comprises both tolerant and intolerant aphids.
Similar with fecundity. Imidacloprid or other insects perhaps weed out the genetically less fecund aphids (although unsure about genetic diversity in these aphids).
Response 14: Excellent comments. This is very enlightening for our next work. There is an objective possibility that the reviewer' questions or concerns are normal; Imidacloprid (pesticide) stress may affect insects in many ways. Imidacloprid LC15 was screened for 2-3 generations in the treatment group and control group (the initial genetic background of insect origin was the same) in a short period of time. The genetic diversity of the test insects may have an effect. However, due to the low concentration and short time factors, we believe that the impact of insecticides on genetic diversity changes in the experimental population is not significant based on no difference between the control and G3. Pesticide stress may have a more direct inductive effect on insect metabolic detoxification related genes in a short time.
Note: In addition to the questions raised by editors and reviewers, we are also revising other questions. As the first author, I apologize for some confusions in our manuscript because I should have checked the manuscript more carefully before it was submitted.
- We have modified the latin from Harmonia axyridis to Hippodamia variegata (line 75-76).
- We have modified the number from 90% confidence intervals to 95% confidence intervals (line 146).
- We have modified the number from 99% to 1 (line 292).
- A 33rd reference is added to the sentence (line430-431).
- Other modifications can be found in the paper (in red font).
Reviewer 2 Report
Journal: insects-2324675
Title: Evaluation of transgenerational effects of sublethal imidacloprid and
diversity of symbiotic bacteria on Acyrthosiphon gossypii
Authors: YinDi Wei, Yue Su, Xu Han, Yue Zhu, Weifeng Guo, Yongsheng Yao *
Submitted to section: Insect Systematics, Phylogeny and Evolution,…
Purpose:
The secondary symbiont Hamiltonella plays an important role in the wheat aphid Sitobion miscanthi and increases its resistance to acetamiprid, chlorpyrifos, cyanobenzamide, and imidacloprid[33]. Together, these studies suggest that symbiotic bacteria mediate insecticide resistance in in sects. Nevertheless, how sublethal concentrations of imidacloprid exposure can affect the biological traits of the population and the symbiotic bacteria in A. gossypii is still unclear.
Life table was constructed to evaluate the sublethal effects of imidacloprid on the biological traits and population growth parameters of A. gossypii. In addition, to explore the potential association of sublethal effects of imidacloprid on, 16S rDNA sequencing analysis was used to study the changes in symbiotic bacteria in the A. gossypii population under imidacloprid stress. The results of this study provide a reference for understanding the chronic effect of imidacloprid stress on the symbiotic bacteria of A. gossypii.
Abstract Symbiotic bacteria and hormesis in aphids are the driving forces
for pesticide resistance. However, the mechanism remains unclear. Here, we
studied the adaptability of Acyrthosiphon gossypii to imidacloprid by
analysing the sublethal effects of imidacloprid on the population growth
parameters and symbiotic bacterial communities. The bioassay results showed
that imidacloprid had high toxicity to A. gossypii with an LC50 of 1.46
mg·L−1. The fecundity and longevity of the G0 generation of A. gossypii
decreased when exposed to the LC15 of imidacloprid. The net reproductive rate
(R0), intrinsic rate of increase (rm), finite rate of increase (λ), and
total reproductive rate (GRR) of G1 and G2 offspring were significantly
increased, but not of the control or G3. In addition, sequencing data showed
that the symbiotic bacteria of A. gossypii mainly belonged to Proteobacteria,
with a relative abundance of 98.68%. The dominant genera of the symbiotic
bacterial community were Buchnera and Arsenophonus. After treatment with the
LC15 of imidacloprid, the diversity and species number of bacterial
communities of A. gossypii decreased for G1-G3, the abundance of
Candidatus-Hamiltonella decreased, but Buchnera increased. These results
provides insight on for the resistance mechanism of insecticides and the
stress adaptation between symbiotic bacteria and aphids.
Suggestions:
Lines 120 and 129: add to Line 129 that you used 10 (??) replicates per treatment
(Line 120) …300 offspring nymphs
(Line 129) ….no less than 30 new nymphs (<24 h) in each group, with 10 replicates per treatment.
Line 229: it is difficult to understand which lines are which –
I guessed the flat yellow and red lines (until day 25) are survival? and
I guess the curved yellow and red lines are fecundity?
You might rewrite the caption to aid reader or make survival as black lines with either yellow (imidacloprid) or red (control) data dots and fecundity as gray lines with yellow (imidacloprid) or red (control) data dots and update legend
